# Chitosan-Based Biomaterial, Calcium Hydroxide and Chlorhexidine for Potential Use as Intracanal Medication

**DOI:** 10.3390/ma14030488

**Published:** 2021-01-20

**Authors:** Bruna de Siqueira Nunes, Rosana Araújo Rosendo, Abrahão Alves de Oliveira Filho, Marcus Vinícius Lia Fook, Wladymyr Jefferson Bacalhau de Sousa, Rossemberg Cardoso Barbosa, Hermano de Vasconcelos Pina, João Emídio da Silva Neto, Solomon Kweku Sagoe Amoah, Carlos Eduardo Fontana, Carlos Eduardo da Silveira Bueno, Alexandre Sigrist De Martin

**Affiliations:** 1Rua Antônio Mariano de Sousa, 31, Antônio Marinho, São José do Egito 56700-000, PE, Brazil; siqueiranunes98@hotmail.com; 2Unidade Acadêmica de Ciências Biológicas (UACB)—Universidade Federal de Campina Grande (UFCG), Avenida Universitária, s/n, Bairro Santa Cecília, Cx Postal 61, Patos 58708-110, PB, Brazil; cesprodonto@hotmail.com (R.A.R.); abrahao.farm@gmail.com (A.A.d.O.F.); 3Laboratório de Avaliação e Desenvolvimento de Biomateriais do Nordeste—CERTBIO/UFCG, Campina Grande 58429-900, PB, Brazil; marcus.liafook@certbio.ufcg.edu.br (M.V.L.F.); wladymyr.bacalhau@certbio.ufcg.edu.br (W.J.B.d.S.); rcbvet@gmail.com (R.C.B.); hermano.pina@certbio.ufcg.edu.br (H.d.V.P.); joao.emidio@certbio.ufcg.edu.br (J.E.d.S.N.); 4Centro de Ciências da Vida, Programa de Pós Graduação em Ciências da Saúde, PUC-Campinas, Av. John Boyd Dunlop, s/n, Jd. Ipaussurama, Campinas 13060-904, SP, Brazil; ceffontana@hotmail.com; 5Instituto de Pesquisas, Endodontia, São Leopoldo Mandic, Faculdade São Leopoldo Mandic, Rua Dr. José Rocha Junqueira, 13, Ponte Preta, Campinas 13045-755, SP, Brazil; carlosesbueno@terra.com.br (C.E.d.S.B.); a-sigrist@uol.com.br (A.S.D.M.)

**Keywords:** chitosan, biomaterial, endodontics

## Abstract

The objective of this study was to develop a chitosan-based biomaterial with calcium hydroxide and 2% chlorhexidine for intracanal treatment application and, consequently, to diminish the number of microorganisms in the root canal system. The chitosan solution was prepared by dissolving it in 2% and 4% acetic acid (*v*/*v*) for 1 h at room temperature (25 °C) with magnetic agitation (430 rpm). Calcium hydroxide was obtained in two stages: the first was the synthesis of the calcium oxide—CaO, and the second was that of the calcium hydroxide—Ca(OH)_2_. The samples were developed using different concentrations of chitosan, calcium hydroxide, and chlorhexidine 2%. They were codified as Ca(OH)_2_ + Q2% (M1), Ca(OH)_2_ + Q4% (M2), Ca(OH)_2_ + Q2% + CLX (M3), Ca(OH)_2_ + Q4% + CLX (M4), Ca(OH)_2_ + Q2% + PEG (M5), and Ca(OH)_2_ + Q4% + PEG (M6). They were characterized through Fourier transform infrared spectroscopy (FTIR), X-ray diffraction (XRD), and rheological measurement, and the antimicrobial activity was evaluated in vitro. Characteristic absorption bands of the source materials used in this research were observed in the FTIR spectra. The X-ray diffraction technique indicated that the material has a semi-crystalline structure and that the presence of calcium hydroxide made the biomaterial more crystalline. The viscosity measurement showed a pseudoplastic behavior of the studied samples. The microbiologic analysis was positive for all samples tested, with bigger inhibition zones for the samples M3 and M4. As a result, we conclude that the formulation developed based on chitosan is promising and has potential to be an intracanal medication.

## 1. Introduction

Many devices for medical use have been developed throughout the decades to act in the biological human systems. They take the form of drug release devices, implantable materials, artificial organs, and wound dressings. These may be described as biomaterials, as they are in contact with the human biological system and can be classified according to the most used ones: ceramic, metallic, composite, and polymers [1].

The use of natural polymers for the production of biomaterials has been extremely valuable, since they can be found in nature, in addition to their characteristics of biocompatibility and biodegradation. The biopolymer known as chitin is formed by units of 2-acetamide-2-deoxy-D-glucopyranose, which are united by glycosidic linkages. Its use has been documented in many papers in the last few years. The structure of chitin is insoluble in water and in many organic solvents. It is obtained through chemical processes, and it is found in the exoskeleton of crustaceans and fungi [2].

Chitosan is obtained from the deacetylation of chitin [3], and it is a biopolymer that can be molded into many physical forms, such as flakes, microspheres, membranes, nanoparticles, tubes, and wires, among others [4]. Its polymeric chains are made up of hydroxyl and amino groups that provide it with hydrophilicity. It can be used in the form of microparticles, gel, films, as a vehicle for medication release, bandages, and injectable gels [5].

The field of application of chitosan is ample and includes agriculture, water treatment, food industry, cosmetics, as well as biopharmaceutic and biomedical uses [2]. Some of its attributes are of wide interest to the odontological field, such as antioxidant, antimicrobial, anti-inflammatory, and healing properties, as well as the ability to inhibit the formation of biofilm [6,7,8,9,10,11,12]. The fact that chitosan is not soluble in water makes it widely applicable in the field of dentistry materials, and it can be added to restoration materials, endodontic cements, and sealants [6,7]. Its properties can be strengthened with the addition of other materials such as calcium hydroxide and chlorhexidine.

Microorganisms and their products are the main etiological factors of pulps with necrotic tissue and periapic lesions. The use of intracanal medications combined with root canal system instrumentation techniques is capable of exerting a significant effect against the host. Among the intracanal medications used in dentistry are phenolic derivatives, aldehydes, halogens, corticosteroids, bases, or hydroxides—e.g., calcium hydroxide, as well as chlorhexidine [13].

Calcium hydroxide is the intracanal medication of choice. Reports have shown that its mineralizing and antimicrobial effects are related to its chemical dissociation into calcium and hydroxyl ions. Its features have been often investigated, but it has shown little to no effect in the destruction of Enterococcus faecalis microorganisms [14,15].

Chlorhexidine is biocompatible, which, in addition to its exceptional antimicrobial action, means that it can be applied in a wide range of contexts, acting against Gram-positive and Gram-negative bacteria, fungi, yeasts, and against the lipophilic virus [16]. The incorporation of chlorhexidine in dental material has shown that its biocompatibility is promoted according to its concentration [17].

Studies in the field of dentistry, which incorporate chitosan particles in endodontic sealants to provide antimicrobial effects between the sealant interface and the root dentine, have shown good results [18,19].

Information gathered with regard to the incorporation of chitosan with antibiotic pastes has shown positive results, not to mention its use as a carrier of antibiotics in regenerative endodontics [20,21].

Considering the above, the objective of this study was to develop a biomaterial based on chitosan, calcium hydroxide, and chlorhexidine to act as an intracanal medication to aid in the decontamination of the system of root canals, especially in cases in which the microorganisms were not eliminated by the chemical–mechanical preparation, in order to diminish the number of bacteria and increase the chances of success in endodontic treatments.

## 2. Materials and Methods

### 2.1. Materials

Low molecular weight chitosan, with a deacetylation degree of approximately 90%, produced in the CERTBIO—UFCG (Campina Grande, PB, Brazil).Glacial Acetic Acid P.A., Sigma Aldrich^®^ (Suzano, SP, Brazil).Calcium hydroxide produced at CERTBIO—UFCG.Polyethylene glycol 400, (PEG). Vetec (São Paulo, SP, Brazil).Chlorhexidine 2%, (CLX) Rioquímica (São José do Rio Preto, SP, Brazil).

### 2.2. Methods

#### 2.2.1. Preparation of the Chitosan Solution

The chitosan solution that was applied was prepared by dissolving 2 g of chitosan in 100 mL of a solution with 2% (*v*/*v*) glacial acetic acid (Q2) and another solution using 4 g of chitosan in 100 mL of a solution with 4% (*v*/*v*) glacial acetic acid (Q4), using mechanical agitation for 1 h at 430 rpm. The pH of the chitosan solutions was verified to be at 2% and 4%, 4.18 and 4.05, respectively, in pH/meter [22,23].

#### 2.2.2. Calcium Hydroxide Synthesis—Ca(OH)_2_

The process of calcium hydroxide synthesis was carried out in two stages. The first was obtaining CaO, for which the CaCO_3_ (P.A. ACS) was used.

Stage 1: Obtaining CaO.
CaCO_3_ = CaO + CO_2_

Initially, CaCO_3_ was thermally decomposed into CaO. The CaCO_3_ was treated at 900 °C for an interval of 2 h with a heat increase of 10 °C/min and subsequent slow cooling. The second stage was obtaining Ca(OH)_2_.

Stage 2: Obtaining Ca (OH)_2_.
CaO + H_2_O = Ca(OH)_2_(1)

At this stage, the CaO was hydrated with ultra-pure water to obtain Ca(OH)_2_. Then, 8 mL of H_2_O was added to each 2 g of CaO, until it became homogeneous. Later, the paste was dried at 100 °C for 5 h, and, finally, the Ca(OH)_2_ was obtained [24].

#### 2.2.3. Sample Development

The biomaterial samples were obtained using the chitosan solutions prepared in “Section 2.2.1”, and adding 0.5 g of the calcium hydroxide as well as 1% chlorhexidine (CLX) or 1% PEG. The substances were manually mixed, and 6 formulations were obtained, which were codified as follows: M1: Ca(OH_2_) + Q2%; M2: Ca(OH_2_) + Q4%; M3: Ca(OH_2_) + Q2% + CLX; M4: Ca(OH)_2_+ Q4% + CLX; M5: Ca(OH_2_) + Q2% + PEG; and M6: Ca(OH_2_) + Q4% + PEG.

### 2.3. Characterization

The samples were characterized using infrared spectroscopy techniques: Fourier transform infrared spectroscopy (FTIR), X-ray diffraction (XRD), rheological measurements, and microbiological evaluation were executed using Staphylococcus aureus and Enterococcus faecalis strains. Figure 1 shows the development flowchart of the samples.

#### 2.3.1. Fourier Transform Infrared Spectroscopy (FTIR)

The Fourier transform infrared spectroscopy (FTIR) was used to identify the possible interactions between the elements that constitute the composites.

The equipment used was a Spectrum 400 spectrometer from Perkin Elmer FTIR/FTNIR (Waltham, MA, USA), with a scan that varies from 4000 to 400 cm^−1^ [25,26].

#### 2.3.2. X-ray Diffraction (XRD)

In this study, the XRD technique was used to identify the crystallinity of the samples, which were then submitted to the X-ray diffraction analysis (XRD) through a Shimadzu X-ray diffractometer (Shimadzu, Quioto, Japan), model XRD 7000, at room temperature, using copper’s Kα radiation (1.5418 Å), with an angular scan of 5° < 2θ < 45°, 40 kV tension, and 30 mA current [27].

#### 2.3.3. Rheological Measurement

Regarding the deformities associated with fluid drainage, the viscosity test was carried out using the rheometer HAAKE ™ MARS ™ (ThermoFisher Scientific, Waltham, MA, USA), at the temperature of 25 °C. The shear rate was varied, changing the speed of the axis that was low for the fluid, and the resistance to torque was converted into shear tension. The viscosity was measured according to a specific shear rate to compare all samples with one another [28].

#### 2.3.4. Microbiological Analysis

##### Method of Agar Disk Diffusion Test

The antimicrobial activity of the samples was investigated, at first, by the method of diffusion in agar described by Andres et al. (2005) and Pelissari et al. (2009). The samples were managed in a sterile environment, using the Quimis biological safety hood (Quimis, Diadema, SP, Brazil), model Q216F21RA1. For the test, the Mueller Hinton (MH) culture medium broth (Kasvi, Italy) was used, to which a solidifying medium (Bacto Agar) was added. The strains used in this study were Gram-positive Staphylococcus aureus (ATCC 25923) and Enterococcus faecalis (ATCC 29212). All strains used showed growth within 24 h. At first, a suspension was prepared in a 0.5 concentration, according to the McFarland scale, using 0.9% sodium chloride for the bacteria. With the aid of a swab, the suspension was spread on Mueller Hinton (MH) agar plates. Then, 30 µL of the positive (NaClO) and negative control (saline solution) was placed on 5 mm-diameter paper-filter disks, which were then placed in MH plates with the bacteria inoculum. The test samples with approximately 5 mm of diameter were directly placed on the MH plates with the bacterial inoculum. The plates were incubated in a bacteriological 35 °C heating chamber, and their diameter were measured after the formation of an inhibition zone. The results were measured using the mean of the three inhibition zones, which were measured 24 h later without subtracting the diameter of the discs. The samples whose inhibition zones grew less than 10 mm were classified as inactive, while those whose halos varied from 10 to 15 mm were classified as active, and those with results above 15 mm were classified as highly active [29,30,31].

## 3. Results and Discussion

### 3.1. Fourier Transform Infrared Spectroscopy (FTIR)

Figure 2a shows the calcium hydroxide spectrum with the absorption band at 3642 cm^−1^ as the stretch of O–H. The absorption bands at 1464, 1080, and 873 cm^−1^ are attributed to different modes of C–O vibration of carbonated groups [32].

In the CLX spectrum (Figure 2b), a band can be noticed in the region between 3700 and 2800 cm^−1^, which refers to the stretch vibrations of amino groups (N–H) present in the structure, as well as to the superposition of the band that refers to the hydroxyl (O–H) group, which refers to water, since it is in a water-based solution. C=C vibrations of the aromatic rings and the stretching band of the C–H which make up the CLX were also present in the spectrum, but they were superposed in this same region. In the region from 1630 and 1650 cm^−1^, a band could be observed that corresponded to the stretching vibration band of the C=N group, in addition to folds of the amino and hydroxyl groups [33]. From 1100 to 1000 cm^−1^, bands related to the Ar-CI (chlorine attached to the aromatic ring) could be observed. It was also possible to observe bands in the CLX spectrum in the region from 800 to 850 cm^−1^ region, resulting from para-disubstituted aromatic rings, which were also present in the CLX structure, corroborating literature [32]. In Figure 2c, related to the sample of the plasticizer (PEG 400) [34], a stretch in the 3441 cm^−1^ band could be observed, which was equivalent to the O-H group; in the bands 2878 cm^−1^, 1464 cm^−1^, and 1343 cm^−1^, are C–H bonds. At 1279 cm^−1^, the groups of OH and C–O–H stand out in a stretch of the band. The same is true for 1094 cm^−1^, approximately, thus corroborating the results found by Kamyar Shameli et al. (2012).

In Figure 2d, the absorption band in 3500 cm^−1^ represents the bond –NH with the stretch of the OH group, which is present in chitosan. The bands at 1406 cm^−1^ and 1379 cm^−1^ are characteristics of the vibration of symmetrical angular deformation of the group –CH_3_, which belongs to the acetamide group. Absorption bands in the 1145 cm^−1^ region correspond to the vibration of the stretch of the groups –C–O–C [34]. Figure 2e shows the same groups and characteristic bands, since this is the same substance (chitosan), and the only difference is its concentration.

According to the spectra of the Ca(OH)_2_ + Q2% and Ca(OH)_2_ + Q4%, in Figure 3a,c, respectively, the absorption bands that are present in calcium hydroxide and those in the 2% and 4% chitosan can be observed. A thin 3642 cm^−1^ band was attributed to the vibration of the free hydroxyl groups. The bands at 1464 and 873 cm^−1^ were attributed to the C–O vibration, carbonate bond. In Figure 3c, related to the Ca(OH)_2_ + Q4% spectrum, a smaller stretch in the chemical bonding band C–O–C also stands out. It is a characteristic of the interaction of hydroxide and the 4% chitosan solution, thus being in accordance with the literature [35].

Figure 3b,d show the biomaterial with the addition of chlorhexidine, showing the presence of absorption bands in the region of 1464 cm^−1^ and 1080 to 873 cm^−1^, corresponding to the band of the bond of carbonate C–O, which is characteristic of the calcium hydroxide, showing that the band was more intense in the 2% composite of chitosan when compared to the 4% one. Furthermore, it was also shown that the chemical grouping band Ar–CI, which is characteristic of chlorhexidine, was present in a stretch of Figure 3b, indicating that both samples showed this stretch in their bands. When the FTIR spectrum of the biomaterials of Figure 3b,d—Ca(OH)_2_ + Q2% + CLX and Ca(OH)_2_ + Q4% + CLX is observed, they are found to have features of the spectra of their precursors of Figure 2a,b,d,e (Q2%, Q4%, Ca(OH)_2_, and CLX) [32].

In Figure 3e,f, the bands characteristic of calcium hydroxide and chitosan stand out, in addition to PEG 400 bands, when they are added to the samples. There was a stretch in the hydroxyl group at 3642 cm^−1^ in the band, which is characteristic of the calcium hydroxyl, in addition to a diminution from 3500 to 3725 cm^−1^ (–OH) of the band. The C–O bond, attributed to the modes of vibration of the carbonate groups, diminished [36].

### 3.2. X-ray Diffraction (XRD)

Chitosan has typical spikes of semi-crystalline materials, with a large base band of approximately 2θ = 20°, with another in 2θ = 10°. This corroborates the results found by Dallan (2005) [37] Xu et al. (2005) [38], Lima (2010) [39], Luo et al. (2011) [40], Cruz (2015) [41], and Rosendo (2016) [42] (Figure 4a).

Calcium hydroxide, on the other hand, has spikes typical of crystalline materials (Figure 4d), corroborating the results found by M. Khachani (2014) [43].

According to the diffractograms in Figure 4c,d, samples that were developed by mixing calcium hydroxide—Ca(OH)_2_ with chitosan 2% and 4% solutions showed the same spikes as in the diffractogram of calcium hydroxide (Figure 4d) [35,36,38]. However, with the addition of the plasticizer, PEG 400, the biomaterial lost crystallinity (Figure 4e,f) [38]. In the samples in which chlorhexidine was incorporated (Figure 4g,h), the calcium hydroxide spikes were also maintained at 2θ = 20° and diminished at 2θ = 10°. There was a predominance of spikes characteristic of calcium hydroxide in all samples evaluated. Regarding chlorhexidine and PEG 400, no significant changes in the diffractograms were found [39,40,41,42,43,44].

### 3.3. Rheological Measurement

The rheological measurements included determinations of apparent viscosity (η) with regard to shear rates (‘γ), as shown in Figure 5.

For an intracanal medication to be considered ideal for clinical application, it should be easily introduced into the root canal in order to ensure adequate contact with tissues. The physical property evaluated in this research (viscosity) affects the handling and application of the medication inside the root canal system, thus justifying the need for this evaluation.

According to Figure 5, it was found that all composites present pseudoplastic behavior; that is, their apparent viscosity diminished as the shear increased [45].

According to the analysis, M1 samples—chitosan 2% + Ca(OH)_2_—and M2 samples—chitosan 4% + Ca(OH)_2_—presented similar viscosities at the start of the shear rate evaluated, showing the same value at the end of the assessment [26].

The M5 sample, 2% chitosan + Ca(OH)_2_ with PEG 400 in its composition, had a higher viscosity at the beginning of the rate analyzed when compared to the M6 sample, which had 4% chitosan + Ca(OH)_2_ associated with PEG 400. However, this rate diminished throughout the analysis [26].

The M4 sample—4% chitosan + Ca(OH)_2_ with chlorhexidine—had a higher viscosity than M3—2% chitosan + Ca(OH)_2_ with chlorhexidine—at the beginning of the analysis. Moreover, both reached the same value of approximately 90–100 γ (1/s) shear rate, when compared to the control group. It was also found that the control group shows higher viscosity in nearly all the shear rates analyzed when compared to the other samples [26].

### 3.4. Microbiological Analysis

Table 1 shows the dimension of the inhibition zones according to microorganism strain and samples. All samples were active with regard to the formation of a zone of inhibition, but the samples M3 (Ca(OH)_2_ + Q2% + CLX) and M4 (Ca(OH)_2_ + Q4% + CLX) had the best results [46,47].

The use of intracanal medications is recommended in cases where the endodontic treatment cannot be concluded in a single consultation, and, as a result, medication is needed to restrict bacterial recontamination. The search for a better alternative of intracanal medication led us to prepare chitosan solution formulations at different concentrations, which is associated with calcium hydroxide and chlorhexidine as antimicrobials, as well as the use of polyethylene glycol as vehicle in some of them.

Calcium hydroxide acts by damaging bacterial DNA, releasing calcium and hydroxyl ions in the periapical medium, providing an alkaline environment, pH 13, which creates an interaction with the microorganism’s cell wall, thus disabling bacterial endotoxins [48]. This medication shows significant results in the elimination of bacteria within one week [49].

Hence, calcium hydroxide—Ca(OH)_2_—is one of the intracanal medications of choice. However, there are some limitations to its use, such as the fact that it is not entirely effective in the elimination of some pathogenic agents of the *Enterococcus* genus, since these agents are resistant to alkaline and can survive in pH = 9 and/or pH = 10 [48].

According to the literature, studies reveal that E. faecalis is the most frequent species found in the root canals of teeth that undergo endodontic treatments. The presence of this species is intimately connected to cases of chronic periapical disease and unsuccessful endodontic treatments. The species E. faecalis has the ability to adapt to severe environmental changes, such as an extreme alkaline pH, salt concentrations, nutrition deprivation, and antimicrobial resistance, and it grows in the root canal in the form of a biofilm [38].

Chlorhexidine is used as an intracanal medication due to its alkaline pH, which gives it antimicrobial characteristics. Its bactericidal effect results from the neutralization of the substrates of microorganisms present inside the root canals, with the continuous release of hydroxyl ions. Chlorhexidine at high concentrations is a bactericidal antimicrobial, but at lower concentrations, it becomes bacteriostatic [31,50]. It has been demonstrated that the application of chlorhexidine in root canals for 7 days brings significant results regarding the reduction of bacteria [51]. When used as an intracanal medicine, chlorhexidine is more effective than calcium hydroxide against E. faecalis infection in the dentin tubules [28,50].

The M3 sample (Ca(OH)_2_ + Q2% + CLX) had a higher antimicrobial activity against Staphylococcus aureus in the first 24 h when compared to Enterococcus faecalis (Figure 6).

The M4 sample (Ca(OH)_2_ + Q4%+ CLX) also had a good result in dealing with the Enterococcus faecalis in the first 24 h; it was found to be very active against both strains used in the study (Figure 7).

On the surface of the bacteria, there are negative charges that can bind to the protonated charges of chitosan, which leads to electrostatic interactions that lead to cellular death [46]. Some factors can influence the antimicrobial activity of chitosan, such as weight, deacetylation degree, solubility, pH of the solution, temperature, and viscosity. In this study, the biggest inhibition zone found for Enterococcus faecalis was in the M3 sample (Ca(OH)_2_ + Q2%+ CLX2%), showing that the concentration of chitosan might have given it the better antimicrobial results [48], especially when compared with the M4 sample, which has the same chemical components but an increase in the chitosan concentration.

In this present study, the developed samples that contain chitosan, calcium hydroxide, and chlorhexidine (M3 and M4) were samples that showed the best antimicrobial activities and can be justified because of the synergistic effect created by the association of the properties of each ingredient.

## 4. Conclusions

Based on the results, biomaterials made of chitosan, calcium hydroxide, PEG-400, and chlorhexidine were developed.

The FTIR showed absorption bands that were characteristic of the source materials used in the research. The X-ray diffraction technique showed that the material has a semi-crystalline structure, and the presence of calcium hydroxide increased its crystallinity. In the viscosity assay, the samples showed pseudoplastic behavior. The microbiological analysis was positive for all samples tested, indicating antibacterial activity against the strains studied, with better antibacterial activities for the M3 and M4 samples due to the synergistic effect of chitosan, calcium hydroxide, and chlorhexidine. As a result, we can conclude that the biomaterial obtained is promising and has potential to be used as an intracanal medication. Among the advantages presented by our proposed product in relation to those that exist on the market, the possibility of developing a new biomaterial with a Brazilian national technology, with materials (calcium hydroxide and chitosan) produced in the Northeastern Laboratory of Evaluation and Development of Biomaterials (CERTBIO)—UFCG, constitutes a renewable and low-cost product.

## Figures and Tables

**Figure 1 materials-14-00488-f001:**
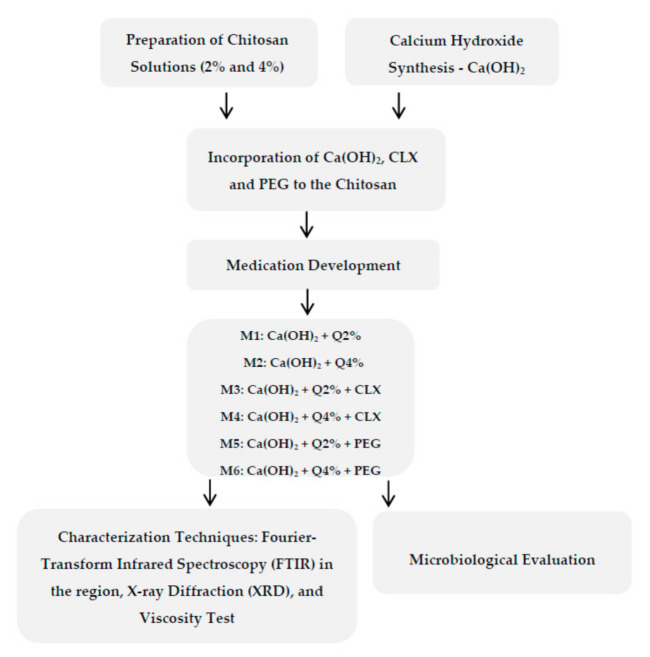
Flowchart of the development of the samples and of the tests conducted.

**Figure 2 materials-14-00488-f002:**
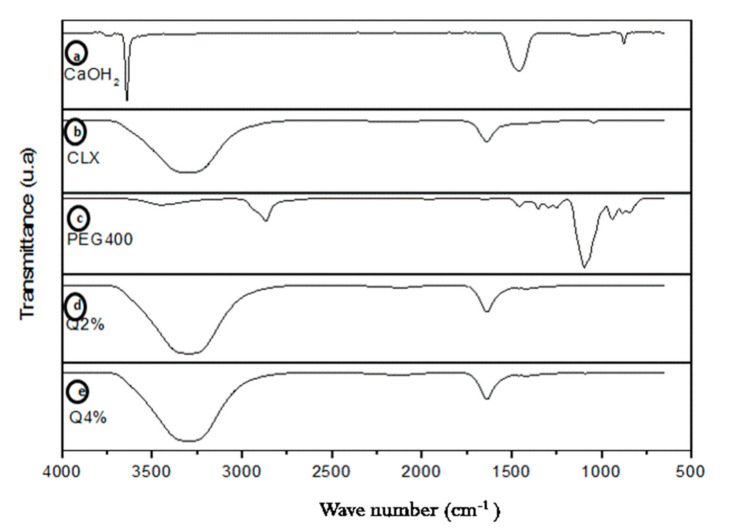
Fourier transform infrared spectroscopy of the Ca(OH)_2_ (**a**); chlorhexidine (CLX) (**b**); polyethylene glycol 400 (PEG 400) (**c**); Q2% (**d**); Q4% (**e**). Q2: 2 g of chitosan in 100 mL of a solution with 2% (*v*/*v*) glacial acetic acid, Q4: 4 g of chitosan in 100 mL of a solution with 4% (*v*/*v*) glacial acetic acid.

**Figure 3 materials-14-00488-f003:**
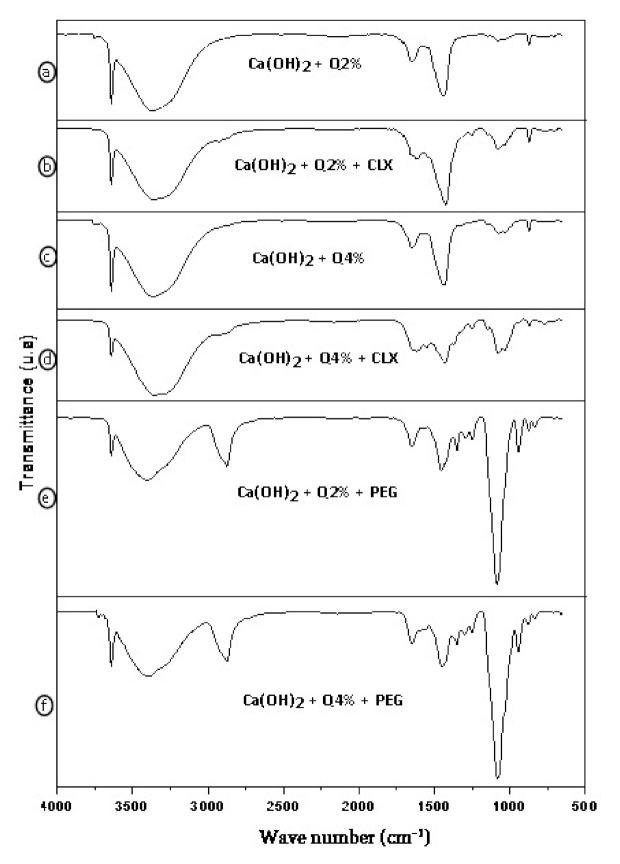
Fourier transform infrared spectroscopy of Ca(OH)_2_ + Q2% (**a**); Ca(OH)_2_ + Q2% + CLX (**b**); Ca(OH)_2_ + Q4% (**c**); Ca(OH)_2_ + Q4% + CLX (**d**); Ca(OH)_2_ + Q2% + PEG (**e**); Ca(OH)_2_ + Q4%+ PEG (**f**).

**Figure 4 materials-14-00488-f004:**
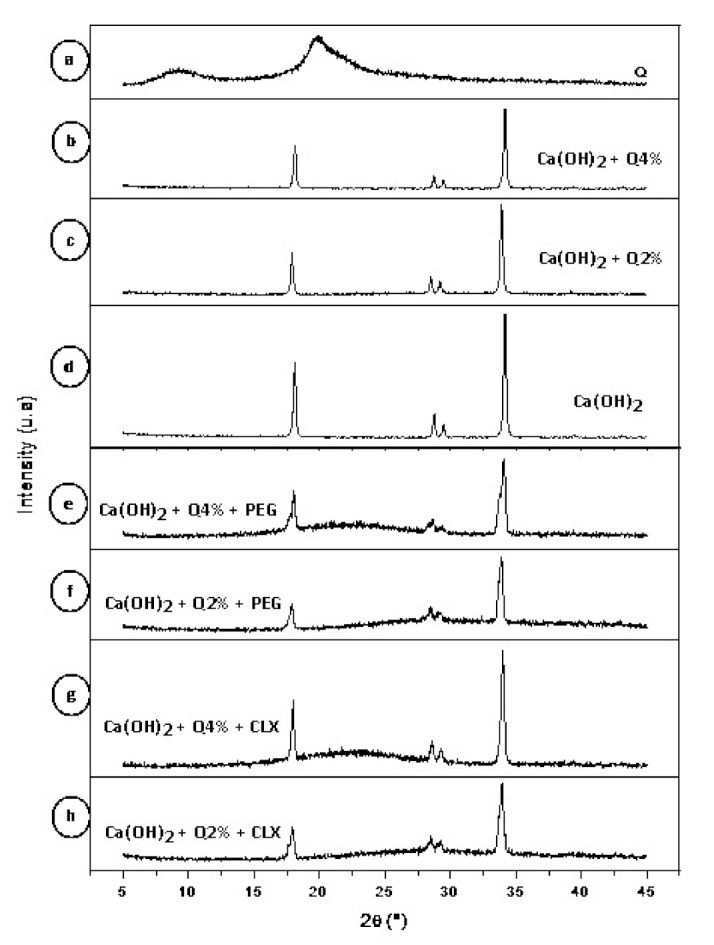
Diffractogram of chitosan powder (**a**); Ca(OH)_2_ + Q4% (**b**); Ca(OH)_2_ + Q2% (**c**); Ca(OH)_2_ (**d**); Ca(OH)_2_ + Q4% + PEG (**e**); Ca(OH)_2_ + Q2% + PEG (**f**); Ca(OH)_2_ + Q4% + CLX (**g**); Ca(OH)_2_ + Q2%+ CLX (**h**).

**Figure 5 materials-14-00488-f005:**
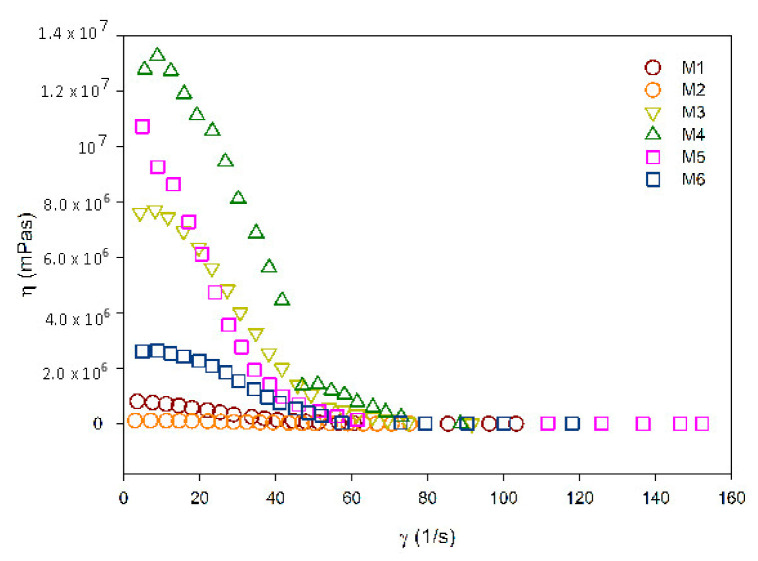
Apparent viscosity curves x shear rates of the samples under study.

**Figure 6 materials-14-00488-f006:**
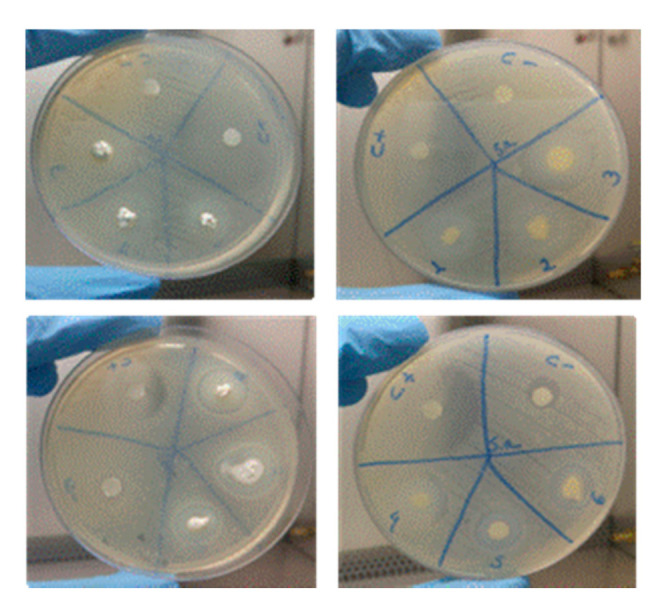
Images of the antimicrobial activity of the samples. Sample 1: Ca(OH)_2_ + Q2%; Sample 2: Ca(OH)_2_ + Q4; Sample 3: Ca(OH)_2_ + Q2% + CLX2%; Sample 4: Ca(OH)_2_ + Q4% + CLX2%; Sample 5: Ca(OH)_2_ + Q4%+ PEG; Sample 6: Ca(OH)_2_ + Q4%+ PEG, all used to deal with the S. aureus ATCC 25,923 strain.

**Figure 7 materials-14-00488-f007:**
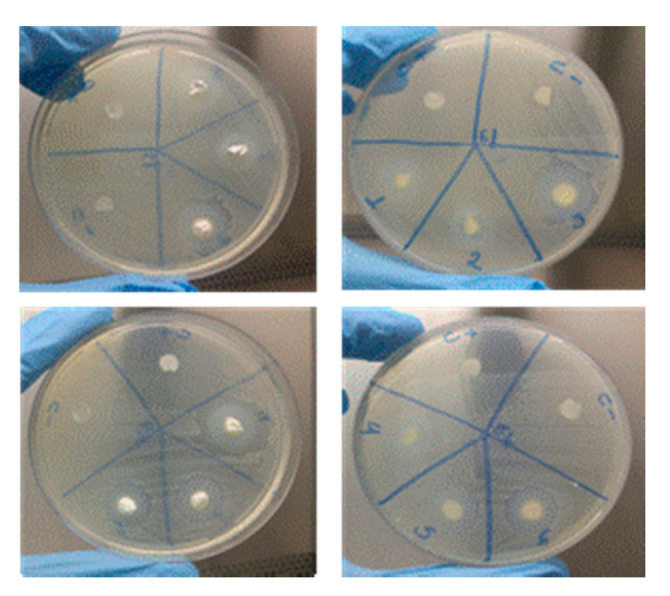
Ca(OH)_2_ + Q2%; Sample 2: Ca(OH)_2_ + Q4; Sample 3: Ca(OH)_2_ + Q2%+ CLX2%; Sample 4: Ca(OH)_2_ + Q4%+ CLX2%; Sample 5: Ca(OH)_2_ + Q4%+ PEG; Sample 6: Ca(OH)_2_ + Q4% + PEG, all used to deal with the *E. faecalis* ATCC 29,212 strain.

**Table 1 materials-14-00488-t001:** Antimicrobial action assay: dimension of inhibition zones according to strain and samples.

Sample	Staphylococcus Aureus ATCC 25923	Enterococcus Faecalis ATCC 29212
	Diameter (mm) 24 h	Diameter (mm) 24 h
M1: Ca(OH)_2_ +Q2%	10	11
M2: Ca(OH)_2_ +Q4%	11	10
M3: Ca(OH)_2_ +Q2%+ CLX2%	22	19
M4: Ca(OH)_2_ +Q4%+ CLX2%	17	20
M5: Ca(OH)_2_ +Q2%+ PEG	17	16
M6: Ca(OH)_2_ +Q4%+ PEG	14	13
(C+) NaClO 2.5%	28	22

## Data Availability

The data presented in this study are available on request from the corresponding author.

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
