# Peer review of "Chitosan-Based Biomaterial, Calcium Hydroxide and Chlorhexidine for Potential Use as Intracanal Medication"

_materials, 2021, doi:10.3390/ma14030488_

Round 1

Reviewer 1 Report

The manuscript from Dr. Siqueira Nunes et.al, the authors studied the development of a novel biomaterial as an intracanal medication to improve the decontamination of the system of root canals. This novel biomaterial is based on chitosan, calcium hydrochloride and chlorhexidine. The manuscript is clear, well organized and written. 

Minor comments:

Authors should introduce better the current medication options available in the field.

Authors should discuss much more in detail the advantages of their proposed medication versus the current options. Why and how this new biomaterial will bring an advantage to the patient?

Author Response

Response to Reviewer 1 Comments

Point 1: Authors should introduce better the current medication options available in the field.

Response 1: The introduction has been improved adding the current medication options available in the field. “Microorganisms and their products are the main etiological factors of pulps with necrotic tissue and periapic lesions. The use of intracanal medications combined with root canal system instrumentation techniques is capable of exerting a significant effect against the host. Among the intracanal medications used in dentistry are phenolic derivatives, aldehydes, halogens, corticosteroids, bases or hydroxides - e.g. calcium hydroxide, as well as chlorhexidine [13]”. “The incorporation of chlorhexidine in dental material has shown that its biocompatibility is promoted according to its concentration [17]”

Point 2: Authors should discuss much more in detail the advantages of their proposed medication versus the current options. Why and how this new biomaterial will bring an advantage to the patient?

Response 2: The advantages of our proposed medication versus the current options has been discussed and can be found in the microbiological analysis section and the conclusion.

Reviewer 2 Report

The manuscript by Nunes et al. reported the preparation of an antibacterial material based on chitosan. Even though some samples showed remarkable antiabcterial activity, the manuscript must be further improved:

  1. There are in total 16 figures in the manuscript, but the information is very limited. More than 80% of them are FTIR and XRD spectra. The authors must rearrange their data with better presentations, e. g. combine some of them into one figure.
  2. Normally, FTIR spectra are shown in ‘transmission-wavenumber’, please correct them so that readers can compare with other reports.
  3. It is not clear how Ca(OH)2 induces bactria death.
  4. Why the blending of PEG further improved bacterial inhibition for the samples?
  5. Since it is use for intracanal application, how about the biocompatbility of the samples? 
  6. The durablility of the materials for antibacterial activity should also be investigated. 

Author Response

Response to Reviewer 2 Comments

Point 1: There are in total 16 figures in the manuscript, but the information is very limited. More than 80% of them are FTIR and XRD spectra. The authors must rearrange their data with better presentations, e. g. combine some of them into one figure.

Response 1: Thank you very much for your observation. The figures have been rearranged and have been reduced to 7.

Point 2: Normally, FTIR spectra are shown in ‘transmission-wavenumber’, please correct them so that readers can compare with other reports.

Response 2: As requested the FTIR spectra has been shown in transmittance-wavenumber

Point 3: It is not clear how Ca(OH)2 induces bacteria death.

Response 3: Thank you very much for your observation. Calcium hydroxide acts by damaging bacterial DNA, releasing calcium and hydroxyl ions in the periapical medium, providing an alkaline environment, pH 13, which creates an interaction with the microorganism's cell wall, thus disabling bacterial endotoxins [48]. It has been added to the manuscript.

Point 4: Why the blending of PEG further improved bacterial inhibition for the samples?

Response 4: Actually PEG was used as a vehicle in this development. However, PEG is known to have some antibacterial activities.

Point 5: Since it is use for intracanal application, how about the biocompatbility of the samples?

Response 5: Thank you very much for your observation. Since all the ingredients used in the development of the samples already have consolidated biocompatibility studies, which have been demonstrated in literature, we think it is not necessary to do biocompatibility studies for now.  

Point 6: The durablility of the materials for antibacterial activity should also be investigated.

Response 6: Thank you very much for your suggestion. We will consider this in our next studies of optimization of our product.

Reviewer 3 Report

Chitosan based biomaterial and calcium hydroxide  for intracanal application

This paper presents the synthesis and characterization of a new biomaterial for endodontics uses. So, the subject is an interesting one, but there are many aspects which must be improved.

First of all, there many English formulations which are wrong from scientific point of view. A lot of explanations between materials interactions are missing. To present only spectra (FTIR and XRD) with few observations is not enough for a scientific paper. Why the authors have done viscosity measurement? I presume what is a reason, but it must be underline in the manuscript.

 I have done many language corrections in the manuscript text, but I am sure that there are more than I have observed.

The conclusions are also very weaks and must be improved.

Author Response

Response to Reviewer 3 Comments

Point 1: First of all, there many English formulations which are wrong from scientific point of view.

Response 1: Thank you very much for your observations. The relevant wrong English formulations have been corrected/revised as suggested.

Point 2:

A lot of explanations between materials interactions are missing. To present only spectra (FTIR and XRD) with few observations is not enough for a scientific paper.

Response 2: Thank you for your observation. But we decided to highlight only the effects of the ingredients used in the development of the samples.

Point 3: Why the authors have done viscosity measurement? I presume what is a reason, but it must be underline in the manuscript.

Response 3: For an intracanal medication to be considered ideal for clinical application, it should be easily introduced into the root canal in order to ensure adequate contact with tissues. The physical property evaluated in this research (viscosity test) affects the handling and application of the medication inside the root canal system, thus justifying the need for this evaluation. It has been added to the manuscript

Point 4: The conclusions are also very weaks and must be improved

Response 4: The conclusion has been improved

Morevover, all observations suggested on the pdf file has been revised for example:

  1. Viscosity on line 26 has been changed to rheological measurement
  2. The sentence on line 26, 27 and 28 has been changed

However, “Line 244 Figure 14. Apparent viscosity curves x shear rates of the samples under study”, we decided to leave the data in one figure, (now figure 5) because, it demonstrates the differences in viscosity at a glance as caused by the presence or absence of the ingredients.

Line 249 and 250 “It stands out that these values could not remain until the end of the values analysed”- has been deleted from the manuscript.

Round 2

Reviewer 2 Report

The current verision of manuscript can be accepted for publication. 

Reviewer 3 Report

This paper could be published in this revised form.